# Modeling and Control of a Spherical Robot in the CoppeliaSim Simulator

**DOI:** 10.3390/s22166020

**Published:** 2022-08-12

**Authors:** Guelis Montenegro, Roberto Chacón, Ernesto Fabregas, Gonzalo Garcia, Karla Schröder, Alberto Marroquín, Sebastián Dormido-Canto, Gonzalo Farias

**Affiliations:** 1Departamento de Electrotecnia e Informática, Universidad Técnica Federico Santa María, Av. Federico Santa María 6090, Viña del Mar 2520001, Chile; 2Escuela de Ingeniería Eléctrica, Pontificia Universidad Católica de Valparaíso, Av. Brasil 2147, Valparaíso 2362804, Chile; 3Departamento de Informática y Automática, Universidad Nacional de Educación a Distancia, Juan del Rosal 16, 28040 Madrid, Spain; 4Ocean and Mechanical Engineering Department, Florida Atlantic University, 777 Glades Road EW 190, Boca Raton, FL 33431, USA

**Keywords:** spherical robot, model, simulation, CoppeliaSim (V-REP)

## Abstract

This article presents the development of a model of a spherical robot that rolls to move and has a single point of support with the surface. The model was developed in the CoppeliaSim simulator, which is a versatile tool for implementing this kind of experience. The model was tested under several scenarios and control goals (i.e., position control, path-following and formation control) with control strategies such as reinforcement learning, and Villela and IPC algorithms. The results of these approaches were compared using performance indexes to analyze the performance of the model under different scenarios. The model and examples with different control scenarios are available online.

## 1. Introduction

The field of robotics is very extensive with respect to robot design, as it is necessary to investigate and analyse the different types of mechanisms that a robot could eventually integrate. Mobile robots can move around their environment using different mechanisms, such as wheels, caterpillars and others [1,2,3,4]. Mobile robots include robots with a spherical shape, that roll to move, like a football. This type of robot represents a particular challenge because it must roll with all of its components inside [5,6]. Examples of this kind of robot can be found in the literature with different driving/steering mechanisms, including single wheels [7,8], pendulums [9,10,11], and omnidirectional wheel mechanisms [12].

It is well-known that, currently, advanced robots are very expensive and can be exposed to damage during laboratory experimentation. For this reason, simulators are very important in this field. Virtual laboratories using these simulators offer significant benefits for robotics education [13]. Using such virtual laboratories, students can test and gain an understanding of concepts that are not easy to follow in the classroom, at any time and pace, and from anywhere [14,15]. They can also design and test control strategies before implementing them in an actual robot in the laboratory without risk of damage to the physical device.

Currently, there are many simulators for different areas of robotics. For example, Argos [16], Webots [17], RFCSIM [18], and CoppeliaSim (formerly V-REP) [19], to mention only those most used. Some of these platforms have licenses that can be used free of charge for educational purposes. They have competitive functionalities with various components that can interact with each other and can be programmed with different programming languages.

The CoppeliaSim simulator deserves special attention for being one of the most widely used for pedagogical purposes today. This simulator provides a versatile and scalable framework for creating 3D simulations in a relatively short period of time. CoppeliaSim has an integrated development environment (IDE) that is based on a distributed and scripted architecture—each scene object can have a built-in script attached, all operating at the same time, in the form of threads. CoppeliaSim contains a wealth of examples, robot models, sensors, and actuators to create and interact with a virtual world at run-time. New models can be designed and added to CoppeliaSim to implement custom-designed simulation experiments [20,21]. CoppeliaSim has an extensive list of mobile robots, among which there is no spherical robot. Therefore, it would be interesting to add a model of a spherical robot so that it would be available and could be used by the community for experiments.

In our laboratory, we have recently developed a spherical mobile robot that can be easily reproduced with a 3D-printer and some basic electronic components. Our idea was to develop and test the simulation model for this robot, as we did in previous investigations [14,15] with a model of the Khepera IV robot in the CoppeliaSim simulator. In the model, the physical properties and components, such as mass, dimensions, and other variables were carefully taken into account. This model was tested with several control strategies under different scenarios. The results obtained with the model were very similar to those obtained with the actual Khepera IV robot with the same control algorithms and experimental conditions [22]. We expect the resulting new model to be very similar to the physical robot, based on our previous results obtained with the Khepera IV robot.

This paper presents the development and testing of a model of a spherical robot whose movement is based on an internal pendulum. The robot consists of a spherical-shaped cover that protects the pendulum and the internal circuitry and allows it to roll to move from one place to another. This type of morphology has a fundamental advantage in that there is no possibility of the robot tipping over, which gives it certain stability in movement. At the same time, it has certain disadvantages with respect to sliding on the surface and difficulties with the presence of obstacles or irregularities in the terrain [9,12,23].

The main contribution of this article is the modelling and control of a non-linear model of a spherical robot that does not exist in the CoppeliaSim simulator. This is a challenging task due to the complexity of the spherical robot model. The developed model is controlled under different scenarios with several control algorithms implemented by the authors in previous studies, including Villela [24], IPC (integral proportional controller) [25], and reinforcement learning (RL) [26,27]. The experiments undertaken to test the robot model included investigation of position control, path-following and formation control. As a result of this work, the model and some examples are available online for use by the community that works with mobile robots.

The article is organised as follows: Section 2 describes the model of the spherical robot and its design and implementation in CoppeliaSim. Section 3 describes the control laws and experiments implemented with the spherical robot. Section 4 shows the tests performed on position control, path-following and formation control. Finally, Section 5 presents the conclusions and discusses future work.

## 2. Spherical Robot Model

This section presents the details of the mathematical model of the spherical robot in different situations.

### 2.1. Robot Description

A spherical robot is a mobile machine that has a spherical casing with a single contact with the surface on which it rests. The housing not only allows the robot to be in a balanced state, but also enables the robot to roll from one place to another without sliding. Figure 1 shows a picture of the actual robot in the laboratory.

For practical purposes, the mechanism that moves the robot and maintains balance is based on a pendulum and three motors. One motor operates the pendulum and modifies the centre of gravity of the robot; the other motors are connected to the casing-sphere, which allows its movement using rotation. The design of this robot is based on [28]. Figure 2 shows a schematic view of the robot, where the right side is the front view and the left side is the lateral view. The components are as follows: (1) casing sphere, a 3D-printed shell which is the body of the robot; (2) bearings, the joins of the case with the shaft; (3) pendulum, which is a stick that allows the robot to turn; (4) counterweights, which are two body masses to improve the stability of the robot; (5) circuitry box, is a small space that contains all the electronic components; (6) shaft (fixed axle), which is the join between the circuit box and the case; and (7) articulation, which is the join between the pendulum and the shaft.

### 2.2. Horizontal Motion Equations

A simplified model of the robot can be seen in Figure 3, which is based on [28]. The model considers moments of inertia, the radius of the sphere, masses of the elements that make up the robot, angles, and the direction of linear velocity.

In this model, it is considered that the casing is always in contact with the surface, which allows the robot to roll without slipping. The dynamic model of the spherical robot for a horizontal movement considers the balance of potential energy, kinetic energy and rotational energy, respectively, as can be seen in Equations (Equation 1)–(Equation 3):(1)$$\left\{\begin{array}{c} U_{1}=0\\ U_{2}=-M_{2}ge\cos\left(\theta_{1}+\theta_{2}\right) \end{array}\right.$$
(2)$$\left\{\begin{array}{c} K_{1}=\frac{1}{2}M_{1}{\left(rw_{1}\right)}^{2}\\ K_{2}=\frac{1}{2}M_{2}\left\{\begin{array}{c} {\left(r\omega_{1}-e\cos\left(\theta_{1}+\theta_{2}\right)\left(\omega_{1}+\omega_{2}\right)\right)}^{2} \end{array}\right. \end{array}\right.$$
(3)$$\left\{\begin{array}{c} T_{1}=\frac{1}{2}J_{1}\omega_{1}^{2}\\ T_{2}=\frac{1}{2}J_{2}{\left(\omega_{1}+\omega_{2}\right)}^{2} \end{array}\right.$$
where the main variables are the following:

$U_{1}$: is the potential energy of the spherical housing with respect to the height of its centroid.

$U_{2}$: is the potential energy of the pendulum with respect to the height of the centroid of the spherical casing.

$K_{1}$: is the kinetic energy of the spherical shell.

$K_{2}$: is the kinetic energy of the pendulum.

$T_{1}$: is the rotational energy of the spherical shell.

$T_{2}$: is the rotational energy of the pendulum.

*e*: is the distance between the centroid of the spherical shell and the pendulum.

$M_{1}$: is the mass of the spherical shell.

$M_{2}$: is the mass of the pendulum.

The Lagrange equations are calculated as follows:(4)$$L=K_{1}+K_{2}+T_{1}+T_{2}-U_{1}-U_{2}$$
(5)$$\frac{d}{dt}\left(\frac{\partial L}{\partial \omega_{1}}\right)-\frac{\partial L}{\partial \theta_{1}}=-T+T_{f}$$
(6)$$\frac{d}{dt}\left(\frac{\partial L}{\partial \omega_{2}}\right)-\frac{\partial L}{\partial \theta_{2}}=T$$
where t is the independent variable of time, T is the torque applied between the casing and the circuitry, and $T_{f}$ is the torque that appears with the friction force that occurs between the casing and the ground. Solving the Equations (Equation 5) and (Equation 6), we obtain:(7)$$T=a_{1}\left(J_{2}-M_{2}re\cos\left(\theta_{1}+\theta_{2}\right)+M_{2}e^{2}\right)+a_{2}\left(J_{2}+M_{2}e^{2}\right)+M_{2}ge\sin\left(\theta_{1}+\theta_{2}\right)$$

This last equation is useful to determine which motor may be used for the construction of the spherical robot and to consider its design.

### 2.3. Robot Turning Equations

The motion that allows changing the direction of the spherical robot is based on the mass of the pendulum using a CMG (control moment gyroscope). The calculation of this motion is based on a torsion pendulum, as shown in Figure 4.

Where $T_{p}$ is the torque of the pendulum; $R_{p}$ is the radius of the pendulum; $\omega_{p}$ is the angular velocity of the pendulum; $A_{c}$ is the swing acceleration of the pendulum; and $M_{2}$ is the mass of the pendulum. The equation of the rotational motion of the pendulum is as follows:(8)$$T_{p}=4R_{p}M_{2}A_{c}$$

Figure 5 shows an angle θ corresponding to the angular position of a spherical mobile robot, whose orientation is controlled by varying the speed of a reaction wheel. The angular velocity of the reaction wheel, relative to the spherical mobile robot, can be varied by varying the voltage applied to the electric motor. This means that, if the motor rotates clockwise, the spherical robot will orient itself in the opposite direction. The effect is achieved by analysing the angular momentum at the axis of rotation; as the speed of the wheel varies, the speed of the mobile robot will also start to vary so that the momentum remains constant.

The equations governing this phenomenon are obtained after analysing the momentum of the spherical robot and the reaction wheel around the axis of rotation. As shown below:(9)$$\dot{\theta }=\omega$$
(10)$$\dot{\omega }=-\frac{B_{1}}{J_{1}}\omega +\frac{B_{2}}{J_{1}}\Omega -\frac{1}{J_{1}}\tau_{m}$$
(11)$$\dot{\Omega }=-\frac{B_{2}}{J_{eq}}\Omega +\frac{1}{J_{eq}}\tau_{m}$$
where the main variables are the following: θ is the angle of the spherical robot casing (robot angle); ω is the angular velocity of the spherical robot; Ω is the angular velocity of the pendulum wheel; $J_{1}$ is the moment of inertia of the spherical robot casing; $J_{2}$ is the moment of inertia of the pendulum wheel of the spherical robot; and $J_{eq}$ is the equivalent moment of inertia where: $\frac{1}{J_{eq}}=\frac{1}{J_{1}}+\frac{1}{J_{2}}$.

### 2.4. Building the Model of the Robot

The parts of the spherical robot were designed using the 3D design software Autodesk Fusion 360 [29] based on the actual robot shown in Figure 2. These parts were then imported into the CoppeliaSim simulator [30] working environment and the robot was assembled manually. Figure 6 shows the process of building the robot.

As can be seen, on the right side, the imported parts of the robot are shown (i.e., housing, pendulum motor, housing motors). On the left side, the result of the robot assembly is shown. The total diameter of the robot is 18 centimetres. Note that the motors and internal elements were built in the same software.

## 3. Experiments with the Spherical Robot

In this section, some tests/experiments implemented with the robot are presented and commented on.

### 3.1. Position Control

This experiment consisted of getting the robot to move from one point C (current position) to another $T_{p}$ (target point) in the most efficient way possible, which implies that it does so by following the shortest path to the destination point. Note that this robot can rotate without displacement, which means that its model is holonomic. To add complexity to the experiment, we imposed non-holonomic constraints on the model. This means that the robot has to move in order to rotate (it cannot rotate about its own position). Figure 7 shows a representation of this experiment.

As can be seen, the variables involved in this experiment were, on one hand, the pose *C*(x,y,θ), which includes the position (*x*,*y*) and the orientation angle (θ) of the robot; on the other hand, the distance (*d*) and the angle (α) at which the target point is located. These variables are calculated as follows:(12)$$d=\sqrt{{\left(y_{p}-y_{c}\right)}^{2}+{\left(x_{p}-x_{c}\right)}^{2}}$$
(13)$$\alpha =\tan^{-1}\left(\frac{y_{p}-y_{c}}{x_{p}-x_{c}}\right)$$

The control law is calculated using the angular error as input ($\alpha_{e}=\theta -\alpha$) and obtaining as outputs the linear velocity (ν) and the angular velocity of the robot (ω). Then the corresponding values for the housing motor and pendulum are calculated for the robot to move with these angular and linear velocities. As a result, the robot is positioned in a new pose C (*x*,*y*,θ), which is used to recalculate the values described above. Figure 8 shows the block diagram of the control loop for this experiment. Where the block Compute implements Equations (Equation 12) and (Equation 13); while the Control Law block can be implemented in different ways, with artificial intelligence or conventional control law approaches, as is explained in the next subsections.

In the case of the spherical robot, the linear velocity (ν) is applied as a voltage to the servomotors of Figure 2, which allows movement of the pendulum backwards or forwards to change the centre of gravity of the robot and to make the robot move in one of those directions. The angular velocity (ω) is applied to the DC motor of the pendulum, which allows the robot to rotate clockwise or counter-clockwise.

In addition to this experiment, the model was tested with other approaches: (1) path-following and (2) formation control. In the first case, the robot must control its position by following a trajectory received as a reference. In the second experiment, more robots were added to the scenario. One of them acts as the leader and the rest as followers. The followers use the position of the leader to make a formation around the leader.

### 3.2. Reinforcement Learning Approach

Reinforcement learning is able to provide an optimal solution despite the complexity of the system. The system learns by acting on the environment while operating in real-time. This is an advantage of traditional optimization methods that rely on a mathematical model and are tuned backwards in time [31,32].

The reinforcement learning approach for this research, called Q-learning, is based on solving the Bellman equation, and the principle of optimality. This technique allows an optimal learning process to be carried out during regular operation, based on the robot’s dynamics, and continuous-time signals. In the limit, the Q-matrix captures a discretized version of the optimal action-state combination in terms of the highest long-term reward.

Given a system described by the dynamics $x_{k+1}=f\left(x_{k},u_{k}\right)$ and a reward function $\sigma (x_{k},u_{k})$, where $x_{k}$ is the state of the system, and $u_{k}=\pi \left(x_{k}\right)$ the control policy, a long-term reward can be defined by Equation (Equation 14):(14)$$\sum_{k=0}^{\infty }\gamma^{k}\sigma \left(x_{k},u_{k}\right)=\sum_{k=0}^{\infty }\gamma^{k}\sigma \left(x_{k},\pi \left(x_{k}\right)\right)$$
where 0<γ<1 is a discount factor required to penalize future rewards and to ensure convergence of the summation. This expression represents the discounted accumulated rewards starting from the current state $x_{0}$ and the application of the policy π.

To apply Bellman’s optimality principle, the previous long-term reward expression (Equation 14) is redefined in terms of the function $Q(x_{k},u_{k})$, called action-value, which allows for the splitting of the reward assignment into two consecutive steps. This action-value function conveys the long-term reward by the contribution of the immediate reward due to applying an arbitrary action $u_{k}$ while in the state $x_{k}$, and by the discounted accumulated reward continuing with the control policy π. This is as shown in (Equation 15) starting from $x_{0}$:(15)$$Q^{\pi }\left(x_{0},u_{0}\right)=\sigma \left(x_{0},u_{0}\right)+\sum_{k=1}^{\infty }\gamma^{k}\sigma \left(x_{k},\pi \left(x_{k}\right)\right)=\sigma \left(x_{0},u_{0}\right)+\gamma \sum_{k=0}^{\infty }\gamma^{k}\sigma \left(x_{k+1},\pi \left(x_{k+1}\right)\right)$$

The optimal value is obtained by maximizing the future rewards; using the optimal policy defined by $\pi^{*}$, a recursive equation is obtained:(16)$$Q^{*}\left(x_{k},u_{k}\right)=\sigma \left(x_{k},u_{k}\right)+\gamma \max_{\mu }Q^{*}\left(x_{k+1},\mu \right)$$

This equation captures the optimal principle by stating that future optimal control actions are not specified by past optimal values, but, instead, only by the current state. The major advance in these calculations is the viability of forward-in-time learning, as opposed to a standard optimal search performed backwards-in-time. This method is also known as Q-learning. From (Equation 16), the following recursive equation can be devised that asymptotically converges to the fixed manifold $Q^{*}$ [33,34]:(17)$$Q^{i+1}\left(x_{k},u_{k}\right)=Q^{i}\left(x_{k},u_{k}\right)+\alpha (\sigma \left(x_{k},u_{k}\right)+\gamma \max_{\mu }Q^{i}\left(x_{k+1},\mu \right)-Q^{i}\left(x_{k},u_{k}\right)).$$

The term $\sigma \left(x_{k},u_{k}\right)+\gamma \max_{\mu }Q^{i}\left(x_{k+1},\mu \right)-Q^{i}\left(x_{k},u_{k}\right)$ is typically labeled temporal difference $TD^{i}\left(x_{k},u_{k}\right)$, or error between the target value $\sigma \left(x_{k},u_{k}\right)+\gamma \max_{\mu }x_{k+1},\mu$ and the current value $Q^{i+1}\left(x_{k},u_{k}\right)$, with 0<α<1 a learning rate. The expression (Equation 17) resembles a gradient descend numerical search. Another interpretation of (Equation 17) is the structure of a low-pass filter, by rearranging it as $Q^{i+1}\left(x_{k},u_{k}\right)=\alpha TD^{i}\left(x_{k},u_{k}\right)+\left(1-\alpha \right)Q^{i}\left(x_{k},u_{k}\right)$. The learning rate α, or numerical search step size, establishes the effect of new information overriding previous information. A small value will reduce the rate of learning, while a larger value will rely more heavily on new data, despite what was previously learned.

### 3.3. Control Laws: Villela and IPC Approaches

As was mentioned before, the Control Law block of Figure 9 can be implemented with traditional control laws or with a machine learning approach. In this subsection, we show both control laws that will later be implemented in the robot. For example, the Villela control law [24], named after its author, was used previously with different kinds of robots with good results [15,18,22]. It calculates the linear velocity (ν) and the angular velocity (ω) of the robot, as shown in Equations (Equation 18).
(18)$$\begin{array}{c} \nu =\left\{\begin{array}{cc} \nu_{max} & if\left|d\right|>k_{r}\\ d\left(\frac{\nu_{max}}{k_{r}}\right) & if\left|d\right|\leq k_{r} \end{array}\right.\\ \omega =\omega_{max}\sin\left(\alpha_{e}\right) \end{array}$$
where $\nu_{max}$ is the maximum linear velocity, $k_{r}$ is the radius of a docking area (around the target point) and $\omega_{max}$ is the maximum angular velocity of the robot.

Based on the Villela control law, in a previous study, we developed what we term an integral proportional controller IPC [25], which was compared with the Villela algorithm and was found to produce better results. The controller implements the velocities as follows in Equation (Equation 19):(19)$$\begin{array}{c} v=\min\left\{K_{v}p\left(\alpha_{e}\right)d,v_{\max}\right\}\\ \omega =K_{p}\sin\left(\alpha_{e}\right)+K_{i}\int_{0}^{t}\alpha_{e}dt \end{array}$$
where $p\left(\alpha_{e}\right)=1-\left|\alpha_{e}\right|/\pi$, for $\alpha_{e}\in \left[-\pi ,\pi \right]$ and $K_{v}$, $K_{p}$, and $K_{i}$ are tuning parameters of the control law. We tested this control law with a differential wheeled mobile robot, so it is challenging to implement this controller with a spherical robot.

## 4. Results

In this section, the results of the implementation of the control experiments with the developed model are presented.

### 4.1. Reinforcement Learning Results

This subsection shows the simulation results for different tests in several iterations to build the Q-matrix. The Q-matrix was obtained in MATLAB during the learning stage and exported to Python (Spyder-Anaconda IDE). The CoppeliaSim software was connected to Spyder via remote API, which ensured it was compatible with Python programming. The experiments were performed with the CoppeliaSim simulator using the developed spherical robot model.

In the learning stage, the algorithm builds the Q-matrix to learn how to reach the destination point. To this end, the angle error ($\alpha_{e}$) is used to obtain the angular velocity (ω) in order to control the position of the robot. Note that, initially, the linear velocity (ν) is kept constant at its maximum value until the robot reaches the docking area.

The Q-matrix is composed of the sets (state, action), where the state is the angle error ($\alpha_{e}$), and the action is the angular velocity (ω). The criterion for obtaining the rewards of the Q-matrix is to penalize significant changes in the angle error and small changes in the angular velocity of the pendulum. In this case, the matrix Q has a size of 126 × 41, where 126 states are generated linearly spaced between −π and π, and 41 actions linearly spaced between −π/2 and π/2. The array is made up of initial reward values. These initial values are adjusted according to the number of iterations of the algorithm based on a learning rate, a discount rate and a coefficient of relationship between exploration and use. They explore and use values to allow the robot to explore the space to complete knowledge of it and later use that knowledge.

Figure 9 shows the results of the position control experiment for different iterations of the RL algorithm (RL 500 m-500.000 iterations, RL 1M-1.000.000 iterations, and so on). The lines describe the trajectories followed by the robot for each value of iteration. The initial position of the robot is represented by the base of the red arrow at (0;0), and the target point is represented by the red cross located at (5;0). The direction of the arrow represents the initial orientation of the robot.

Figure 10 shows the distance to the destination point for these experiments. The y-axis represents the distance in meters and the x-axis represents the time in seconds. As can be seen, for all experiences, the time to arrive at the destination was similar, around 14 s. This would be expected given the similarity between the trajectories shown in the figure above.

The quality of each control algorithm can be evaluated using performance indexes. These indexes use the integral of the error, which is, in our case, the distance to the target point. The performance indexes considered in this work are the following: (1) integral square error (ISE), (2) integral absolute error (IAE), (3) integral time squared error (ITSE), and (4) integral time absolute error (ITAE). Note that the last two also include the time in the analysis [35]. Table 1 shows the performance indexes to compare the results of each algorithm. Note that all the indexes showed similar results, which is logical in view of the above results. The best performance was shown by the RL4M algorithm. For that reason, we selected this algorithm to compare with the other approaches.

### 4.2. Comparison between Different Approaches (RL, Villela and IPC)

To establish a basis for comparison of the results of the different control algorithms with the spherical robot, in addition to the RL, the Villela and the IPC algorithms were selected. In both algorithms, the parameters were selected based on our previous experience with the implementation of these experiments with the Khepera IV robot (see for example [25,27]). In the Villela algorithm, the parameters were the following: $V_{max}=1$ and $\omega_{max}=\pi /2$. For the IPC algorithm, the parameters were the following: $K_{v}=0.15$, $K_{p}=1.5$, $K_{i}=0.000001$, $V_{max}=1$ and $\omega_{max}=\pi /2$.

Figure 11 shows the results of the position control experiment for the different algorithms (Villela, IPC, and RL). As in the previous case, the red arrow represents the initial orientation of the robot and the red cross represents the target point. The lines describe the trajectories followed by the robot for each control law. The initial position of the robot is represented by the base of the arrow at (0;0), and the target point located at (5;0).

As can be seen, the IPC and RL4M algorithms describe similar trajectories, while Villela’s approach shows the worst trajectory. In order to provide a better comparison, we can analyze the graph of distance vs. time. Figure 12 shows the distance to the destination point for these experiments. The y-axis represents the distance in meters and the x-axis represents the time in seconds.

As can be seen, the better performance was demonstrated by the RL4M algorithm, which took around 14 s to reach the destination point. Note that, in the previous figure, it appears that IPC had a similar trajectory to RL4M, but when the time is taken into account in the analysis, the differences are clearer. With the IPC algorithm, the robot took more than 20 s to reach the destination point. So the trajectory was similar but took more time and the performance was the worst of all, while RL4M showed the best behaviour.

Table 2 shows the performance indexes for all algorithms. As can be seen, as was expected, the best performance was shown by the RL4M algorithm, which confirms the previous results.

### 4.3. Path Following

To test the control strategies in a different scenario, we implemented a path-following example [36,37,38]. This experiment is widely known in the field of mobile robot control because it is used to demonstrate the behaviour of the implemented control algorithm. It consists of “dynamic” position control of the robot in which the reference point constantly changes to describe a trajectory by joining all points. The result is that the robot follows the points one by one to create the trajectory. Figure 13 shows the implementation of this experiment with the spherical robot in CoppeliaSim.

Figure 14 shows the trajectories described by the robot for different control algorithms: RL4M (red line), Villela (green line) and IPC (violet line). The dashed line represents the trajectory that the robot receives as a reference. As can be seen, the robot follows the trajectory with different behaviours for all the algorithms.

At first glance, it appears that the best performance was shown by the Villela algorithm. To perform a better comparison, we calculated the performance indexes for all algorithms. Table 3 shows these results.

As can be seen, the lowest values in all indexes were for the Villela algorithm, which means that, for this algorithm, the robot followed the trajectory better.

### 4.4. Multi-Agent Formation Control

This experiment was based on [22,39] and consisted of making a formation in a cooperative and decentralized way. One robot acted as the leader and the rest as followers. The positions of the follower’s robots were controlled as in the previous experiment. To make a formation, the followers’ robots have to reach a position using the leader position as the reference. Equation (Equation 20) shows how the velocity of the leader robot is calculated as a function of its own position error ($E_{pm}$) and the followers’ errors in the formation ($E_{f}$).
(20)$$\nu_{m}\left(t\right)=K_{p}E_{pm}\left(t\right)-K_{f}E_{f}\left(t\right)$$

The values of $K_{p}$ and $K_{f}$ are manually adjusted to control the influence of each error in the velocity of the leader robot. If $K_{f}=0$, the errors of the followers in the formation are not taken into account, and the control is made in a non-cooperative way because the leader robot does not consider the errors of the followers. Equation (Equation 21) shows how the formation error is calculated.
(21)$$E_{f}\left(t\right)=\sum_{i=1}^{N}E_{pi}\left(t\right)$$

Figure 15 shows this experiment in the CoppeliaSim simulator for the RL algorithm. For the leader robot, the reference is the target point at the left of the image (red semi-sphere) and, for the followers, the target points are their positions in the formation. In this case, the followers use the position of the leader to make a triangular formation around it. Both followers are situated at a fixed 4 m from the leader and $30^{\circ }$ and −$30^{\circ }$ behind it, respectively.

As can be seen, initially, the robots make a triangle using the leader robot as a reference. At the end of the experiment, the followers maintain the formation around the leader robot. Figure 16 shows the data for this experiment. The blue small circle represents the initial position of the leader robot and the green cross represents the destination point. The blue line represents the trajectory described by the leader robot and the red and orange lines represent the trajectories described by the following robots. As can be seen, the robots maintain the formation during the experiment.

Figure 17 shows the results of this experiment for all the algorithms. The y-axis represents the distance travelled by the robots. The leader robot is represented by the blue lines and they show the distance from the robot to its target point. The followers are represented by the red and orange lines, which show the distance between each follower and the leader.

As can be seen, at the beginning, the leader robot moves away from the target because, initially, the target is at its back. After a few seconds, the leader reaches the destination point, while the followers maintain a constant distance to the leader (4 m), which means that the formation is maintained during the experience.

By simple visual inspection, it can be observed that the RL algorithm showed better performance because the leader robot reached the destination point in less time and travelled the shortest distance. However, to be on the safe side, we calculated the performance indexes for each experiment to establish a more accurate comparison. Table 4, Table 5 and Table 6 show the performance indexes for each algorithm and robot in each experiment.

All experiences were generated with the same initial conditions, and only the control algorithm was changed in each case. We can then compare the results using Equation (Equation 21), which is shown in the row Sum in Table 4, Table 5 and Table 6. As can be seen, the least values in all cases for the Sum row were observed for the RL experiment, which confirms that the better performance was produced by this algorithm.

## 5. Conclusions

This article presents the design and implementation of a model of an actual spherical robot, the method of movement of which is based on an internal pendulum. The design of the model was developed using the 3D-design and modeling software, Autodesk Fusion 360. The model was incorporated piece-by-piece into the CoppeliaSim simulator where the hardware was assembled; the position control strategy was programmed in the LUA and Python programming languages to verify its operation. Different experiments, concerning position control, path-following, and multi-robot formation control were performed. The results obtained with the different control laws and experiments showed that the design and implementation of the robot model were satisfactory since its behaviour was similar to that previously obtained with a differential model of the Khepera IV robot. Future work will include performing these experiments with the actual robot in the platform previously implemented in our laboratory [22]. This is a challenging task due to the complexity of obtaining the absolute position of the robot in the platform.

## Figures and Tables

**Figure 1 sensors-22-06020-f001:**
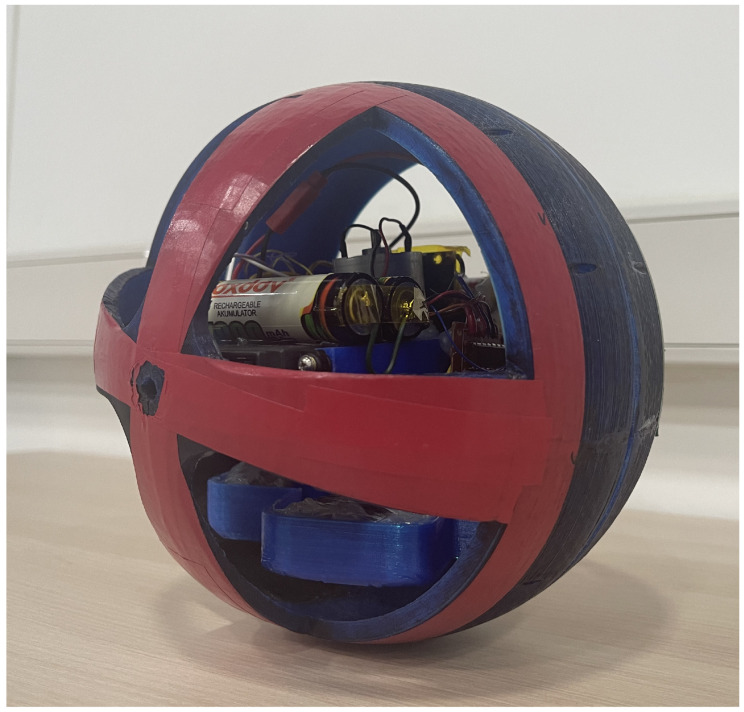
Actual spherical robot.

**Figure 2 sensors-22-06020-f002:**
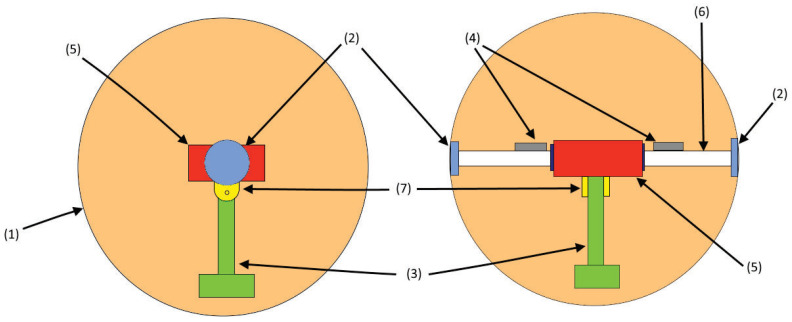
Details of the spherical robot.

**Figure 3 sensors-22-06020-f003:**
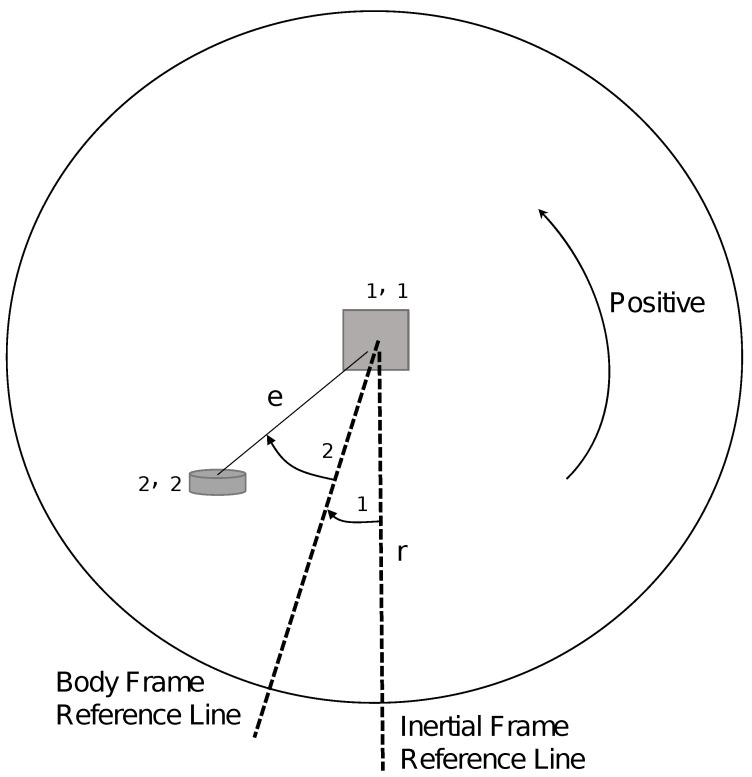
Simplified scheme of the horizontal movement of the robot (1 rotation angle of the ball, 2 rotation angle of the pendulum with respect to the ball).

**Figure 4 sensors-22-06020-f004:**
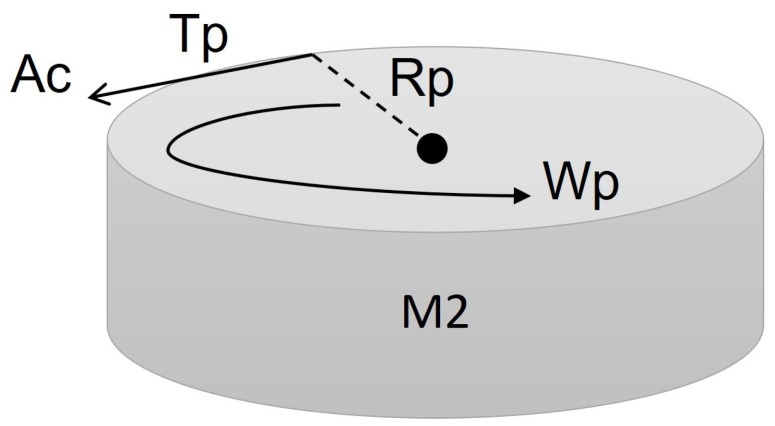
Pendulum schema reaction wheel.

**Figure 5 sensors-22-06020-f005:**
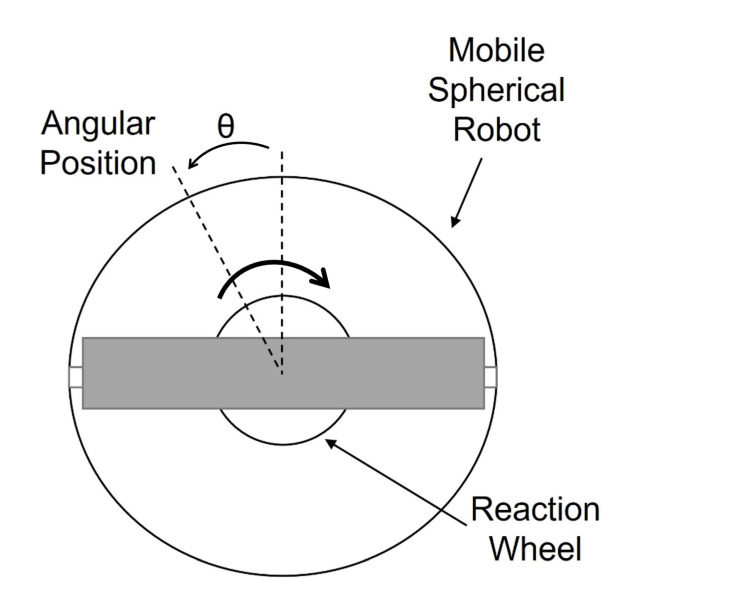
Top view of the robot for rotational movement.

**Figure 6 sensors-22-06020-f006:**
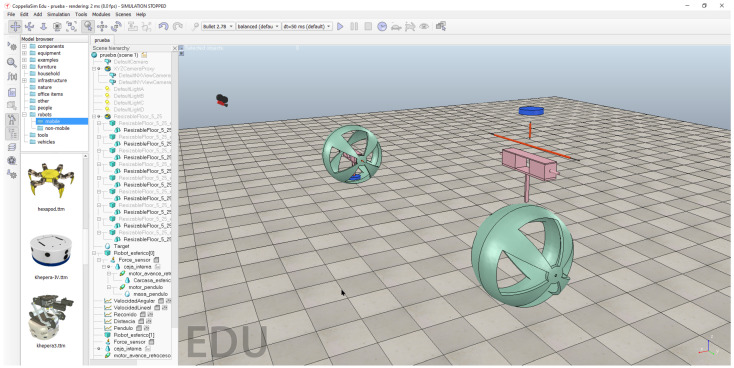
Assembling the robot in CoppeliaSim.

**Figure 7 sensors-22-06020-f007:**
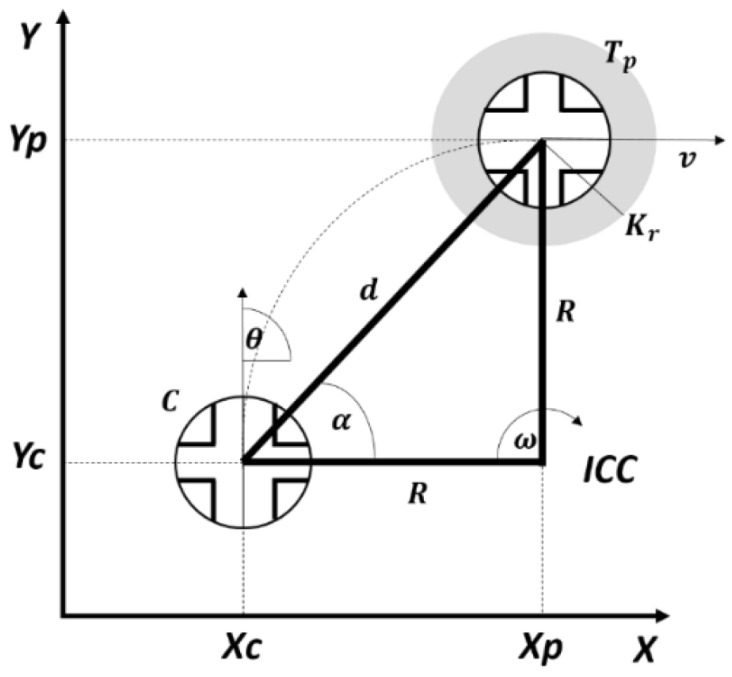
Position control experiment.

**Figure 8 sensors-22-06020-f008:**
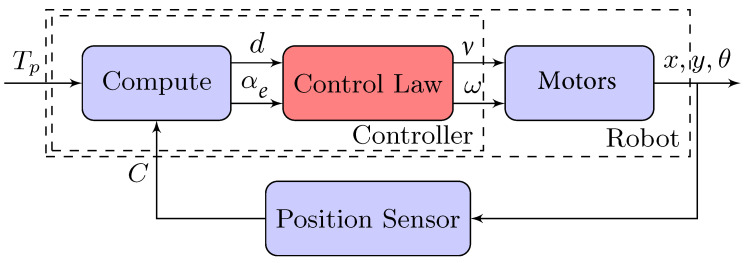
Position control block diagram.

**Figure 9 sensors-22-06020-f009:**
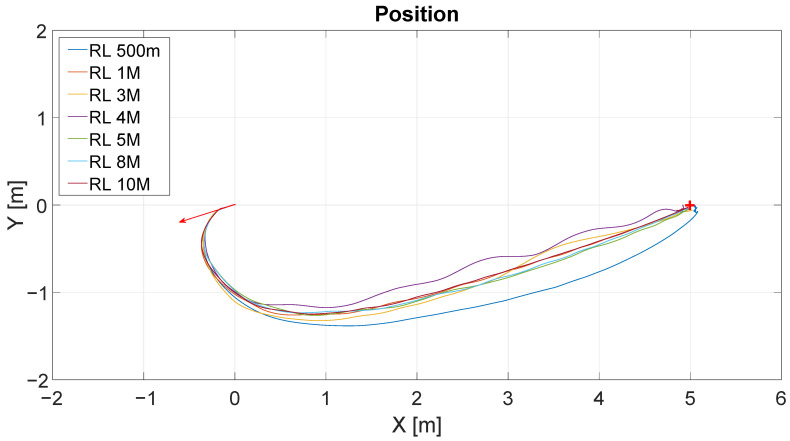
Obtained trajectories for different values of iteration in RL algorithm.

**Figure 10 sensors-22-06020-f010:**
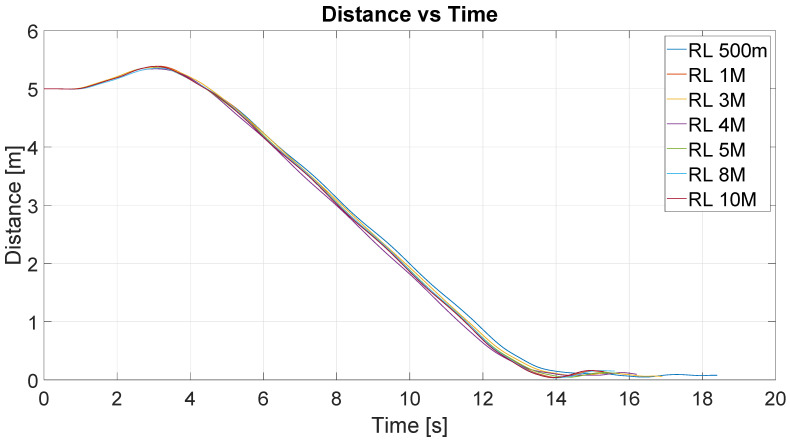
Distance vs. time of all experiences/iterations.

**Figure 11 sensors-22-06020-f011:**
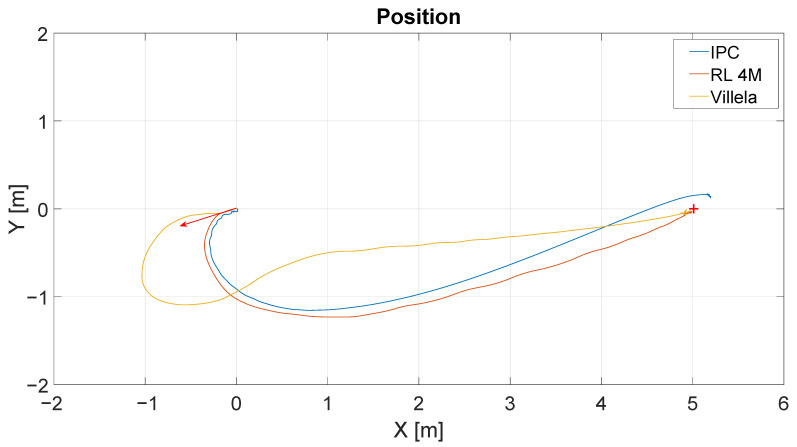
Obtained trajectories for each control algorithm (Villela, IPC and RL4M).

**Figure 12 sensors-22-06020-f012:**
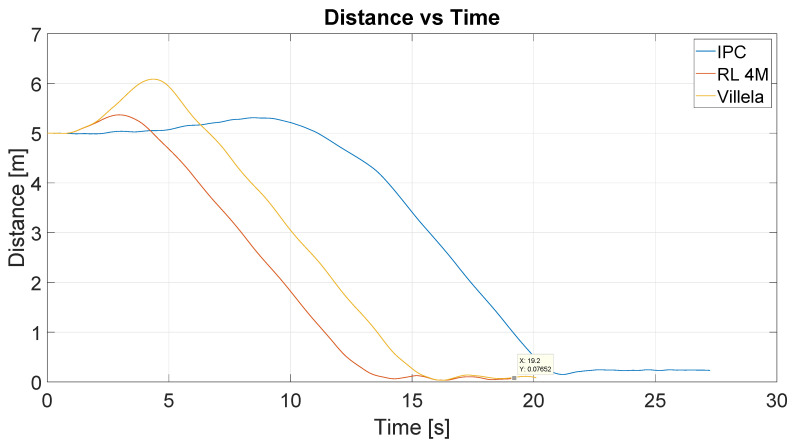
Distance vs. time of all experiences (RL, Villela and IPC).

**Figure 13 sensors-22-06020-f013:**
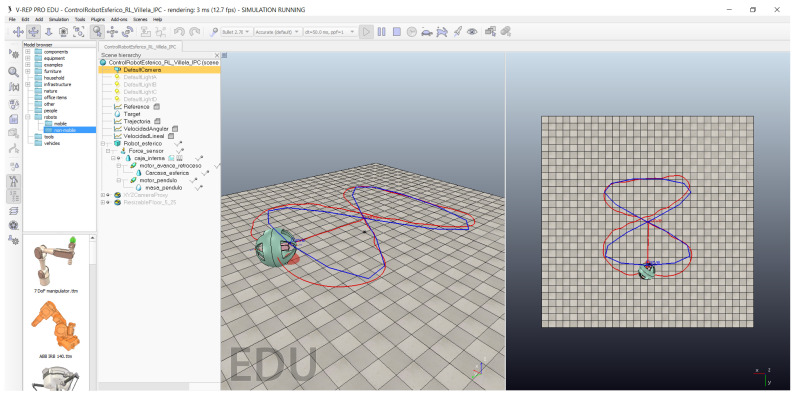
Path following of a Lissajous figure.

**Figure 14 sensors-22-06020-f014:**
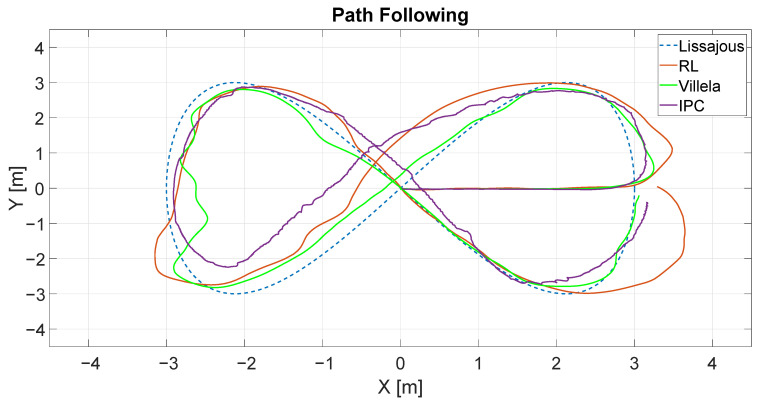
Path following example for Villela, IPC and RL4M.

**Figure 15 sensors-22-06020-f015:**
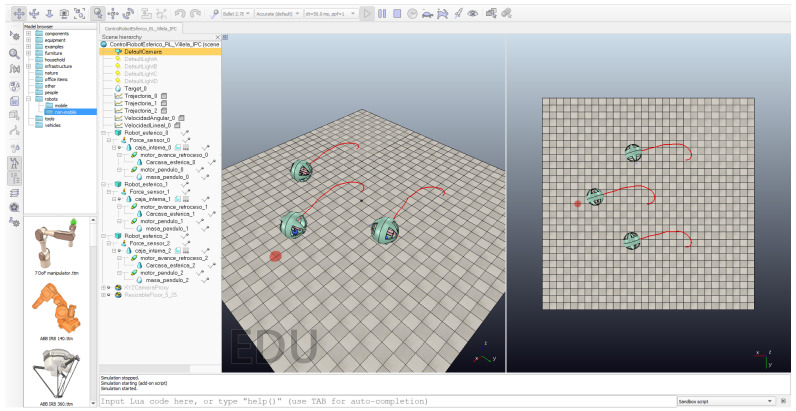
Formation control experiment.

**Figure 16 sensors-22-06020-f016:**
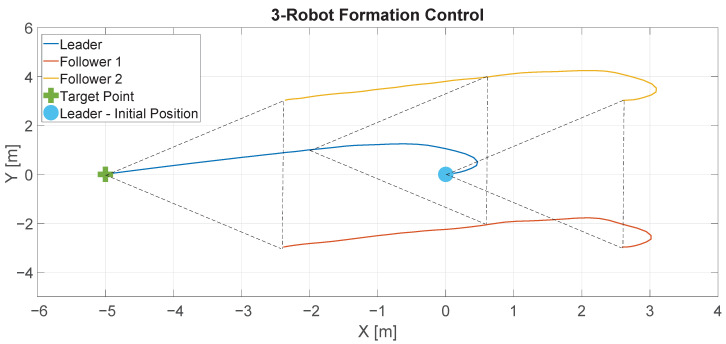
Results of the formation control experiment.

**Figure 17 sensors-22-06020-f017:**
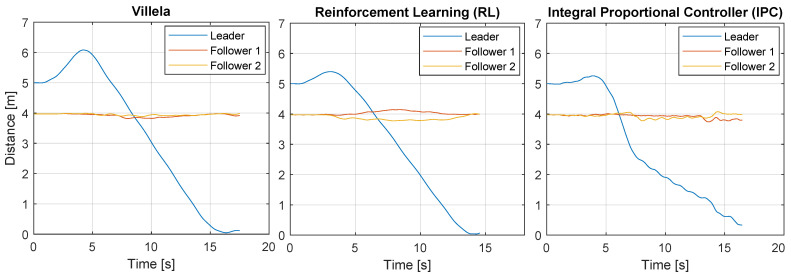
Positions of the robots for all algorithms.

**Table 1 sensors-22-06020-t001:** Performance indexes for each algorithm. In bold, the better case.

Index	RL 500 m	RL 1M	RL 3M	RL4M	RL 5M	RL 8M	RL 10M
IAE	46.81	45.84	46.25	**45.19**	45.67	45.66	45.52
ISE	198.29	195.68	197.05	**191.79**	194.27	194.47	194.23
ITSE	804.22	781.07	791.03	**755.07**	775.13	776.11	771.11
ITAE	234.15	222.85	227.18	**217.66**	221.73	221.49	219.67

**Table 2 sensors-22-06020-t002:** Performance indexes for each algorithm in the position control experiment. In bold, the better case.

Index	Villela	RL4M	IPC
IAE	58.59	**45.39**	85.28
ISE	273.33	**191.93**	388.68
ITSE	1.334.0	**754.55**	2.985.5
ITAE	338.50	**220.96**	743.39

**Table 3 sensors-22-06020-t003:** Performance indexes for each algorithm in the path-following experiment. In bold, the better case.

Index	RL4M	Villela	IPC
IAE	87.79	**85.12**	212.07
ISE	162.91	**162.12**	384.11
ITSE	4.18 × 10${}^{3}$	**3.78 × 10${}^{3}$**	2.34 × 10${}^{4}$
ITAE	2.25 × 10${}^{3}$	**2.02 × 10${}^{3}$**	1.32 × 10${}^{4}$

**Table 4 sensors-22-06020-t004:** Performance indexes for the Villela algorithm in the formation control experiment.

Robot	IAE	ISE	ITSE	ITAE
Leader	57.97	271.45	1.32 × 10${}^{3}$	332.37
Follower 1	68.41	268.24	2.34 × 10${}^{3}$	599.01
Follower 2	68.90	272.06	2.38 × 10${}^{3}$	603.86
Sum	195.28	811.76	6049.30	1535.24

**Table 5 sensors-22-06020-t005:** Performance indexes for the RL4M algorithm in the formation control experiment.

Robot	IAE	ISE	ITSE	ITAE
Leader	46.00	197.80	802.10	224.79
Follower 1	58.41	235.35	1.72 × 10${}^{3}$	427.47
Follower 2	56.12	217.32	1.57 × 10${}^{3}$	408.16
Sum	**160.53**	**650.48**	**4103.30**	**1060.42**

**Table 6 sensors-22-06020-t006:** Performance indexes for the IPC algorithm in the formation control experiment.

Robot	IAE	ISE	ITSE	ITAE
Leader	95.52	377.80	3.11 × 10${}^{3}$	1.04 × 10${}^{3}$
Follower 1	129.12	506.93	8.24 × 10${}^{3}$	2.11 × 10${}^{3}$
Follower 2	129.44	509.42	8.38 × 10${}^{3}$	2.13 × 10${}^{3}$
Sum	354.09	1394.17	19,747.70	5297.60

## Data Availability

Not applicable.

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
