# Peer review of "Modeling and Control of a Spherical Robot in the CoppeliaSim Simulator"

_sensors, 2022, doi:10.3390/s22166020_

Round 1

Reviewer 1 Report

This article presents the design and implementation of a spherical robot model whose movement method is based on an internal pendulum. The model was incorporated in parts into the CoppeliaSim simulator where the hardware was assembled and its position control strategy was programmed in the LUA and Python programming languages to verify its operation.

There are still some issues that need to be addressed.

(1)   The modeling of the spherical robot model is a key step in this paper. However, the author did not intuitively give an introduction to the detailed parameters of this model, such as joint positions and angles. Please provide detailed parameters.

(2)   The parameters of RL algorithm in 4.1 are not equidistant, so it is suggested to supplement the selection basis.

(3)   Section 4.2 describes the comparison of position control results under different control strategies. It is suggested to briefly introduce the parameters selection of Villela algorithm and IPC algorithm.

(4)   For the path following results described in 4.3, it is mentioned that the performance of Villela algorithm is better than other algorithms. But Section 5 only introduces the performance of RL algorithm, and Villela algorithm is not mentioned. It is suggested to supplement the corresponding statement.

(5)   In 4.4, please provide the reasons for comparing various control strategies in formation control simulation.

(6)   There are still some formatting problems, the picture annotation is not directly below the picture, and some paragraphs are not indented with the first line.

Author Response

Comments:

This article presents the design and implementation of a spherical robot model whose movement method is based on an internal pendulum. The model was incorporated in parts into the CoppeliaSim simulator where the hardware was assembled and its position control strategy was programmed in the LUA and Python programming languages to verify its operation.

Authors response:  Thank you for these comments.

There are still some issues that need to be addressed.

1.- The modeling of the spherical robot model is a key step in this paper. However, the author did not intuitively give an introduction to the detailed parameters of this model, such as joint positions and angles. Please provide detailed parameters.

Authors response:  Thank you for this comment. Done. We have modified section 2.1 and replaced figure 2 to explain in more detail the parts of the robot.

2.- The parameters of the RL algorithm in 4.1 are not equidistant, so it is suggested to supplement the selection basis.

Authors response:  Thank you for this comment. Done. We have updated section 4.1 with a paragraph that explains the details of Matrix Q in our experiments.

3.-  Section 4.2 describes the comparison of position control results under different control strategies. It is suggested to briefly introduce the parameters selection of the Villela algorithm and IPC algorithm.  

Authors response:  Thank you for this comment. Done. We added a paragraph to section 4.2 to clarify the parameters selection.

4.- For the path following results described in 4.3, it is mentioned that the performance of the Villela algorithm is better than other algorithms. But Section 5 only introduces the performance of the RL algorithm, and the Villela algorithm is not mentioned. It is suggested to supplement the corresponding statement. 

Authors response:  Thank you very much for this comment. Done. We have modified both the 4.3 and 4.4 Sections to unify the comparison between the algorithms. We also added a comparison using performance indexes for all experiments/algorithms.

5.- In 4.4, please provide the reasons for comparing various control strategies in formation control simulation. 

Authors response:  Thank you for this comment. Done. We have rewritten this subsection to introduce the comparison between the algorithms using performance indexes, tables, and figures.

6.-  There are still some formatting problems, the picture annotation is not directly below the picture, and some paragraphs are not indented with the first line. 

Authors response:  Thank you for this comment. Done. Regarding the alienation of the figures, we had a mistake with the interpretation of the MDPI LaTEX template that we had fixed. Regarding the indentation, this journal allows no indentation when the paragraph after an equation is a continuation of a previous text.

Reviewer 2 Report

The following issues should be considered:

1. The manuscript must be written in accordance with the MDPI style.

2. What kind of perturbations can be considered in this kind of system?

3. The references are out of date. With the aim of highlight the contribution of the paper, recent references must be found related to the state-of-the-art review.

4. English must be improved.

5. Comparison with similar systems reported in literature must be executed.

Author Response

Comments: 

The following issues should be considered:

1.- The manuscript must be written in accordance with the MDPI style.

Authors response:  Thank you for this comment. Done. We have used the MDPI LaTEX template.

2.- What kind of perturbations can be considered in this kind of system?

Authors response:  Thank you for this comment. We think that some obstacles ramps or steps can be added to the environment. Another kind of perturbation can be that the floor has an angle of inclination. The robot could be hit by an object to deflect it from its trajectory.

3.- The references are out of date. With the aim of highlighting the contribution of the paper, recent references must be found related to the state-of-the-art review.

Authors response:  Thank you for this comment. Done. We have added new references with different recent articles on spherical robots.

4.- English must be improved.

Authors response:  Thank you for this comment. We have reviewed the article.

5.- Comparison with similar systems reported in literature must be executed.

Authors response:  Thank you for this comment. Done. We have added a comparison of different control algorithms and we have evaluated the results with performance indexes.

Reviewer 3 Report

Review for the Modeling and control of a spherical robot in the CoppeliaSim
simulator submission. The article is well written, original, addresses a
current topic. The article presents the design, mathematics and
implementation of a spherical robot model, whose movement method is based on
an internal pendulum. It particularly deals with the kinematics of the robot
and the methodology behind the task. It gives a very detailed explanation of
the setup and discusses several test cases.

While the methods are sound, the robustness of the algorithm should be
discussed.

Overall, the scientific contribution of the paper should be identified, since spherical robots have been around for a decade, not only in simulations. In this aspect, the paper could also include some references
like:
-https://www.research-collection.ethz.ch/bitstream/handle/20.500.11850/154271/eth-7943-01.pdf
https://ieeexplore.ieee.org/abstract/document/9512918?casa_token=tKaVj_kcT3o
AAAAA:oLI13DJbCiAFw7UnUFKuwSGCLXxmGplyjzIADMz2YnC0nwqcAMBTaid5pgo1ZI8WmfEDg3
gHpG-4

The paper otherwise could be accepted after minor revision.

Author Response

Comments: 

1.- Review for the Modeling and control of a spherical robot in the CoppeliaSim simulator submission. The article is well written, original, and addresses a current topic. The article presents the design, mathematics, and implementation of a spherical robot model, whose movement method is based on an internal pendulum. It particularly deals with the kinematics of the robot and the methodology behind the task. It gives a very detailed explanation of the setup and discusses several test cases.

Authors response:  Thank you for these comments.

2.- While the methods are sound, the robustness of the algorithm should be discussed. 

Authors response:  Thank you for this comment. Done. RL is by definition a multivariable, nonlinear, optimal controller that learns from the dynamics of the environment, which includes the robot’s dynamics and external dynamics. The robustness of this type of controller lies in its ability to keep its learning process during operation, by using its current signals to keep adjusting its Q-value function. The richer the data gathered from the environment, the more robust the controller will be. Any change in the environment’s dynamics will be eventually acquired by RL adjusting the Q-value function. 

3.- Overall, the scientific contribution of the paper should be identified, since spherical robots have been around for a decade, not only in simulations. In this aspect, the paper could also include some references like: 

- https://www.research-collection.ethz.ch/bitstream/handle/20.500.11850/154271/eth-7943-01.pdf 

-https://ieeexplore.ieee.org/abstract/document/9512918?casa_token=tKaVj_kcT3oAAAAA:oLI13DJbCiAFw7UnUFKuwSGCLXxmGplyjzIADMz2YnC0nwqcAMBTaid5pgo1ZI8WmfEDg3gHpG-4

Authors response:  Thank you for these comments. Done. We have modified the introduction section to clarify the hypothesis and contribution of this research, which are basically:

- Hypothesis: It is possible to develop the non-linear model of a spherical robot that does not currently exist in CoppeliaSim. For this, we base ourselves on our previous work, in which we developed a model for an existing robot (Khepera) that was not incorporated in this simulator either, and the results were significantly good.

- Contribution: the development and testing of a non-linear model of a spherical mobile robot that does not exist in the CoppeliaSim simulator. We think that this model could be very useful for the community due to its complexity in controlling its position.

We have only added the second of the references you suggest because we believe that the rest are not directly related to our research.

4.- The paper otherwise could be accepted after minor revision.

Authors response:  Thank you for this comment.

Round 2

Reviewer 1 Report

The paper is well-revised and can be accepted now.